# The Role of Rumen Microbiota and Its Metabolites in Subacute Ruminal Acidosis (SARA)-Induced Inflammatory Diseases of Ruminants

**DOI:** 10.3390/microorganisms10081495

**Published:** 2022-07-25

**Authors:** Yunhe Fu, Yuhong He, Kaihe Xiang, Caijun Zhao, Zhaoqi He, Min Qiu, Xiaoyu Hu, Naisheng Zhang

**Affiliations:** Department of Clinical Veterinary Medicine, College of Veterinary Medicine, Jilin University, Changchun 130062, China; fuyunhesky@163.com (Y.F.); heyuhong99@sina.com (Y.H.); xiangkaihe@sina.com (K.X.); zhaocj2001@sina.com (C.Z.); hezq199706@163.com (Z.H.); qmin2019@163.com (M.Q.)

**Keywords:** subacute ruminal acidosis, inflammatory diseases, low-grade inflammation, rumen microbiota, metabolites

## Abstract

Subacute ruminal acidosis (SARA) is a common metabolic disease in ruminants. In the early stage of SARA, ruminants do not exhibit obvious clinical symptoms. However, SARA often leads to local inflammatory diseases such as laminitis, mastitis, endometritis and hepatitis. The mechanism by which SARA leads to inflammatory diseases is largely unknown. The gut microbiota is the totality of bacteria, viruses and fungi inhabiting the gastrointestinal tract. Studies have found that the gut microbiota is not only crucial to gastrointestinal health but also involved in a variety of disease processes, including metabolic diseases, autoimmune diseases, tumors and inflammatory diseases. Studies have shown that intestinal bacteria and their metabolites can migrate to extraintestinal distal organs, such as the lung, liver and brain, through endogenous pathways, leading to related diseases. Combined with the literature, we believe that the dysbiosis of the rumen microbiota, the destruction of the rumen barrier and the dysbiosis of liver function in the pathogenesis of SARA lead to the entry of rumen bacteria and/or metabolites into the body through blood or lymphatic circulation and place the body in the “chronic low-grade” inflammatory state. Meanwhile, rumen bacteria and/or their metabolites can also migrate to the mammary gland, uterus and other organs, leading to the occurrence of related inflammatory diseases. The aim of this review is to describe the mechanism by which SARA causes inflammatory diseases to obtain a more comprehensive and profound understanding of SARA and its related inflammatory diseases. Meanwhile, it is also of great significance for the joint prevention and control of diseases.

## 1. The Pathogenesis of Subacute Ruminal Acidosis

Subacute ruminal acidosis (SARA) is a common nutritional metabolic disease in ruminants, especially in high-yield dairy cows. In recent years, to improve the milk production performance of dairy cows, farmers have used a large number of high-grain diets as feed, which has induced a series of metabolic diseases, especially SARA. SARA will reduce milk yield and decrease the fat content of the milk of dairy cows [1]. Meanwhile, it causes a series of diseases, such as diarrhea, mastitis and laminitis, and causes huge economic losses to dairy farming [1,2].

Current definitions of SARA are based on the pH of rumen fluid. Due to the different methods of rumen fluid collection, the pH of rumen fluid that defines SARA is usually between 5.5 and 6.0. Garrett et al. suggested that a pH of 5.5 be used as the threshold of SARA when rumen fluid samples were collected by rumenocentesis [3]. Moreover, Plaizier suggested that a pH of 6.0 be used as the threshold of SARA when rumen fluid samples were collected with a stomach tube [4]. Furthermore, as rumen pH varies considerably throughout the day, the timing after feeding of rumen fluid sampling plays an important role in rumen pH. Therefore, a diagnosis of SARA usually uses a threshold of a rumen pH depression from 5.2–5.8 for at least 3 h/day [5,6].

Ruminal pH is mainly affected by the concentration of organic acids, such as volatile fatty acids (VFAs) and lactic acids, in the rumen. Under a normal fermentation condition, the concentration of VFA in the rumen increases gradually with the utilization of carbohydrates by the microbiota, and the rumen pH gradually decreases. The acidic rumen environment promotes the removal of VFA absorption through the rumen epithelium [7]. In addition, rumination will increase the amount of saliva secretion, which contains many alkaline substances [8]. Thus, the rumen pH is rapidly increased to normal physiological levels by absorption and neutralization of VFA in the rumen, and rumen pH is maintained at normal physiological levels before and after each feeding (Figure 1). 

After feeding, rumen microbes ferment the carbohydrate intake of dairy cows to produce VFA and lactate, which reduces the rumen pH. Part of the temporarily increased VFA is absorbed by the rumen wall and part is neutralized by secreted saliva to keep the rumen pH within a certain range. 

When dairy cows eat a large amount of concentrate feed for a long time, on the one hand, excessive carbohydrates will produce excessive VFA under the fermentation of rumen microbiota; on the other hand, a low proportion of neutral detergent fiber (NDF) cannot stimulate ruminants and does not produce enough saliva to neutralize the large accumulation of VFA in the rumen, resulting in rumen pH drops below the physiological threshold [9,10] (Figure 2). In addition, a large number of studies showed that rumen microbiota and metabolites were significantly changed when SARA occurred in dairy cows [7,11,12]. Recently, research also indicated that responsive changes in the rumen microbiota and metabolome were associated with susceptibility to SARA in dairy cows [13]. These results suggest that the rumen microbiota plays a crucial role in the occurrence of SARA in dairy cows.

Once dairy cows ingest too many carbohydrates, the VFA generated in the rumen and lactic acid energy substances cannot be timely absorbed by the rumen wall, and the amount of saliva secretion is reduced, resulting in a pH lower than 5.6–5.8 in the rumen, and SARA occurs. 

## 2. Relationship between SARA and Its Related Inflammatory Diseases in Ruminants

The peripartum period refers to the period from 3 weeks prepartum to 3 weeks postpartum and is an important period affecting the health of the cow, milk production and the occurrence of many diseases [14]. In recent years, along with the huge demand for milk production and economic efficiency of dairy cows, a higher level of nutrition and feeding management has been necessary in the cow feeding process. At the same time, the incidence of dairy diseases has increased, especially in the periparturient period, when cows experience stress from parturition, lactation, feed conversion and environmental changes. In the periparturient period, metabolic diseases (SARA, ketosis, etc.) and inflammatory diseases (mastitis, endometritis, etc.) have the risk of high incidence, clusters and complications [15,16,17]. SARA is associated with inflammation of different organs and tissues in dairy cows. Studies have shown that SARA can cause diarrhea, rumen mucosal damage, laminitis, mastitis and liver abscesses in dairy cows [18]. Thus, clarifying whether SARA is associated with these diseases and elucidating the mechanisms of the association are essential for the prevention and control of peripartum dairy diseases.

### 2.1. SARA and Liver Disorders

The liver is the final barrier preventing gastrointestinal bacteria and their products, such as Lipopolysaccharide (LPS) and histamine, from entering systemic circulation [19]. Systemic clearance and detoxification of LPS occurs in Kupffer cells in the liver [20]. High concentrations of LPS in the portal and hepatic veins of cows with SARA can be observed. Nutrient metabolism in the liver is inhibited and the immune function of the liver is reduced in the presence of strong endotoxins [20]. Both exogenous and endogenous LPS can induce inflammatory liver injury [21]. Subsequently, excess LPS beyond the metabolic capacity of the liver causes oxidative stress and cellular damage in the liver and leads to the development of inflammation [22,23]. Activation of TLR4 signaling pathways may play an important role in the inflammatory process [24]. Meanwhile, the levels of acute phase proteins (APPs), including haptoglobin (HP) and serum amyloid A (SAA), in plasma and their mRNA expression in the liver were significantly increased, suggesting that animals suffering from SARA experienced a certain stress status [25]. SARA is also capable of causing epithelial damage to the rumen, where pathogenic bacteria enter the liver tissue through the circulating bloodstream and colonize and cause liver abscesses [9]. In addition, the expression levels of genes related to lipid formation were downregulated during SARA, whereas those of catabolic genes and some inflammatory genes were upregulated [26]. These results indicated that SARA could induce inflammation of the liver, liver injury and liver abscesses. 

### 2.2. SARA and Mastitis

Mastitis, the inflammation of the mammary gland, is the most frequent disease of dairy cows. The occurrence of the disease causes a decline in the milk yield and milk quality of dairy cows and brings huge economic losses to dairy farming. A large number of studies have shown a link between SARA and mastitis [2,27]. A recent study we conducted showed that somatic cell count (a well-established direct indicator of mastitis) increased in milk in a high-grain diet-induced SARA model of dairy cows [2,28]. The results suggested that SARA could induce the occurrence of mastitis. Mastitis induced by SARA is caused by endogenous pathogenic microorganisms and/or by endogenous bacterial metabolites from the rumen. Ours and other studies also showed that LPS concentrations in the rumen fluid and lacteal artery and vein increased in dairy cows with SARA induced by high-grain feeding [2,28]. Increased LPS could induce an inflammatory response in the mammary gland [2,28]. Wang et al. also showed that increased rumen iE-DAP was observed in long-term, high concentrate feeding-induced SARA, and increased iE-DAP could activate NOD1-NF-κB signaling pathway-dependent inflammation in the mammary gland of mid-lactating cows [17]. Meanwhile, studies have shown that the blood–milk barrier plays a critical role in the development of mastitis [29]. A recent study we conducted showed that the permeability of the blood–milk barrier increased in SARA [2]. These results indicated that SARA could increase susceptibility to pathogen-induced mastitis. However, the relevant mechanism still needs to be further clarified. 

### 2.3. SARA and Endometritis

Endometritis, inflammation in the inner lining of the uterus, is the major uterine disease of dairy cattle. Under normal conditions, the uterus is exposed to bacteria during calving. A healthy immune system can resist pathogenic colonization. Recent studies have shown that the gut microbiota plays a critical role in the regulation of the host’s immune system [30]. It plays an important role in the regulation of the inflammatory response and in the protection against infections [31]. Studies have demonstrated that gut microbiota dysbiosis can decrease the host’s immunity, increasing inflammation and intestinal and parenteral infections (endometritis and mastitis) as occurs with metabolic diseases. Recent studies showed that SARA could lead to inflammation of the uterus, decrease immunity and increase susceptibility to endometritis [32,33]. Muhammad Shahid Bilal et al. [32] reported that rumen-derived LPS could induce inflammation of the uterus during SARA. In our study, we also showed that gut microbiota dysbiosis could increase inflammation of the uterus [34]. Furthermore, Jeon et al. [35] showed that pathogenic bacteria could migrate from the gut to the uterus through the blood, causing metritis in dairy cows. In our study, we found that gut microbiota dysbiosis could increase the susceptibility to *Staphylococcus aureus*-induced endometritis [34]. These results indicated that SARA could result in inflammation of the uterus and increase susceptibility to endometritis. However, the relevant mechanism still needs to be further clarified. 

### 2.4. SARA and Laminitis

Chronic laminitis is the most important clinical sign in dairy herds suffering from SARA, and once the prevalence of chronic laminitis is above 10%, it indicates the occurrence of SARA in the dairy herd [36]. Acidosis induced by high-grain or high-sugar diets plays an important role in the development of laminitis. It is widely believed that laminitis is a local manifestation of metabolic disorders, with lactic acid, endotoxins and histamine being the main factors that induce laminitis [37,38]. SARA causes the pH of the system to decrease and subsequently activates vascular activity, increasing blood pressure in the vascular system of the hoof. Meanwhile, rumen endotoxins and histamine are released into the bloodstream, increasing vasoconstriction and dilation [39]. Subsequent vascular damage with oedema, dermal congestion and even thrombus formation eventually leads to dilation of the dermis and increasing severity of laminitis [40]. The study found a reduced probability of laminitis in cows after oral administration of a vaccine against LPS, further demonstrating the important role of LPS in the development of laminitis [41]. The critical link between lactic acidosis and laminitis appears to be associated with persistent hypoperfusion, which results in ischemia in the digit [42]. In addition, our recent study found that laminitis in dairy cows was not only associated with elevated levels of lactate and LPS but also closely related to the bacterial community of the rumen, characterized by elevated abundance of bacteria that enrich acid-enhancing metabolites [39].

## 3. Low-Grade Inflammation in SARA

Low-grade inflammation (LGI), also known as subclinical inflammation or low-grade chronic inflammation, is a chronic and low-grade inflammatory pathological state. LGI refers to inflammation below the level of infectious and autoimmune inflammation, without local and systemic symptoms such as redness, swelling, heat and pain, and exhibits a subclinical pathological state that is easily ignored [43]. LGI is associated with an increased risk of ill health, poor well-being and mortality. Additionally, it is associated with an increased risk of several diseases, such as diabetes, Alzheimer’s disease and cancer [44,45,46]. By collecting data from reported SARA cases or high-grain diet-induced SARA models, we found that various inflammatory indices in blood, rumen, feces, tissues and organs were increased (Table 1). This suggests that the body is in a systemic inflammatory state during the pathogenesis of SARA. Because most cases do not exhibit obvious external manifestations or clinical symptoms, they are more in line with the judgement standard of LGI. Therefore, we believe that the animal body is in a low-grade inflammatory state, or “subhealth” state, during the pathogenesis of SARA. The duration of this LGI state often depends on the degree of gastrointestinal mucosal injury, individual differences and the intervention measures taken in the pathogenesis of SARA. 

## 4. The Mechanism of SARA-Mediated IGL and Related Inflammatory Diseases 

Inflammation acts as both a friend and foe. Inflammation is beneficial as an acute, transient reaction to harmful conditions, facilitating the defense, repair, turnover and adaptation of many tissues. However, IGL is likely to be detrimental for many tissues and for normal functions. A large body of studies has demonstrated a significant link between a mild proinflammatory state and many diseases. IGL contributes to the development of many diseases, such as metabolic syndrome (MetS), nonalcoholic fatty liver disease (NAFLD), type 2 diabetes mellitus (T2DM) and osteoarthritis [55,56,57,58]. Meanwhile, studies have shown that these diseases are accompanied by gut microbiota disorder [59,60,61,62].

In recent years, many possible triggers of LGI have been proposed. Among these factors, the gut microbiota has been reported to play a central role in IGL. The role of gut-microbiota-mediated intestinal inflammation has long been appreciated [57]. However, there are an increasing number of studies highlighting that gut bacteria and/or their metabolites may drive the IGL in the host and extraintestinal distal organs. Therefore, combined with the literature, we believe that the dysbiosis of the rumen microbiota, the destruction of the rumen barrier and the dysbiosis of liver function in the pathogenesis of SARA will lead to the entry of rumen bacteria and/or metabolites into the body through blood or lymphatic circulation and keep the body in a chronic low-grade inflammatory state. Meanwhile, rumen bacteria and/or their metabolites can also migrate to the mammary gland, uterus and other organs, affect immune function and increase susceptibility to infectious diseases (Figure 3). 

Rumen microbiota disorder causes the rumen bacteria and their harmful metabolites to be released into the blood, damages liver function and induces a systemic chronic inflammatory response. Then, these harmful substances enter the body through the circulation of the blood and lymph circulation, enter various tissues and organs, increase the risk of metabolic diseases, reduce the defensive capabilities of the tissues and organs and increase the risk of infectious diseases. 

### 4.1. Ruminal Microbiota Dysbiosis in SARA 

The rumen is the largest and most important digestive organ of ruminants. The rumen is similar to an anaerobic biological fermentation tank, which contains a large number of microorganisms, including bacteria, fungi, protozoa, archaea and viruses. These microbiotas have been functionally connected to digestion and absorption, metabolism, immune homeostasis and the neuroendocrine regulation of the host. An increasing number of studies have demonstrated that the rumen microbiota is closely related to the production performance of dairy cows [63,64]. Disturbance in the rumen microbiota is associated with the development of various diseases. Recent studies showed that alterations in rumen microbiota and metabolite activity were observed in dairy cattle and goats with SARA [65,66]. The richness and diversity of the rumen and fecal microbiota were reduced in grain-based SARA of Danish Holstein cows [67]. At the phylum level, *Bacteroidetes* and *Firmicutes* were the most abundant phyla in the rumen. Meanwhile, the abundances of *Firmicutes* and *Actinobacteria* increased, and the abundances of *Cyanobacteria* and *Verrucomicrobia* in the rumen decreased in SARA cows. At the genus level, the abundances of *Rickettsiales*, *Acholeplasmatales*, *Victivallaceae*, *Sutterella* and *Shuttleworthia* decreased, and the abundance of *Succinivibrio* in the rumen increased in SARA cows [68]. Chen et al. [11] identified 527 bacterial genera in the rumen of dairy goats; the relative abundances of *Ruminococcus*, *Candidatus*, *Saccharimonas*, *Lachnospiraceae*, *Eubacterium coprostanoligenes*, *Pseudobutyrivibrio* and *Saccharofermentans* increased, and those of *Prevotella*, *Rikenellaceae*, *Prevotellaceae*, *Succiniclasticum* and *Bacteroidales* decreased during a grain-based SARA challenge. Mao et al. [69] identified 155 different genera in the rumen of dairy cows, of which the relative abundances of *Prevotella*, *Treponema*, *Anaeroplasma*, *Papillibacter* and *Acinetobacter* decreased, and those of *Ruminococcus*, *Atopobium*, unclassified *Clostridiales* and *Bifidobacterium* increased in SARA cows.

Ruminal metabolism is closely related to alterations in the rumen microbiota. Previous studies showed that alterations in rumen metabolite activity were observed in dairy cattle and goats with SARA. Mao et al. [70] reported that bacterial degradation products (xanthine, uracil, hypoxanthine, etc.), inflammatory compounds (LPS, ethanolamine, glutaric acid, lactate, biogenic amines including putrescine, histamine, tryptamine, tyramine, etc.) and amino acids (alanine, glycine, isoleucine, etc.) increased in the rumen of dairy cows during a grain-based SARA challenge. The levels of cinnamic acid, benzoic acid, lactose, dihydroxyacetone, etc., decreased in the rumen of dairy goats during a grain-based SARA challenge. Yang et al. [71] identified that 144 differential metabolites of these 73 metabolites, such as dihydroxyacetone, L-fucose, D-mannose, sucrose, isomaltose, D-lyxose, D-maltose, galactinol, DL-lactate, propionic acid, hypoxanthine, indole3-carboxylic acid, 5-hydroxyindoleacetate, 5-methoxydimethyltryptamine, 3-methylxanthine, acetylmannosamine, picolinic acid and thymine, increased in the rumen fluid when animals were fed a high-corn diet. Burim N. Ametaj et al. [72] reported that N-nitrosodimethylamine, dimethylamine, lysine, leucine, phenylacetylglycine, nicotinate, glycerol, fumarate, butyrate, valine, lactate, LPS, etc., increased in the rumen of dairy cows fed grain-based diets. 

### 4.2. The Destruction of the Rumen Barrier in SARA

A large body of studies have confirmed that gut microbiota dysbiosis can cause intestinal tissue damage and enhanced permeability, resulting in a “leaky gut” [73,74]. Meanwhile, the bacteria and/or their metabolites in the intestinal tract can migrate to extraintestinal distal organs such as the lung, liver and brain through endogenous pathways, leading to related diseases [74,75]. Therefore, maintaining intestinal barrier integrity is essential for the health of humans and animals. Similarly, rumen barrier integrity is important for the health of ruminants. The rumen barriers, constituted by the microbial, physical and immune barrier, prevent the transmission of pathogens and toxins to the host tissue in the maintenance of host-microbe homeostasis. Studies have shown that the ruminal epithelial barrier is important for a healthy and productive cow [76]. However, the destruction of the rumen barrier has been observed in many diseases of ruminants. Previous studies showed that high-grain, diet-induced SARA could lead to the destruction of the rumen barrier, resulting in an increase in ruminal epithelial permeability [2]. Zhang et al. [77] reported that high-concentrate feeding could induce ruminal epithelial inflammation by upregulating inflammation-related gene expression. Zhang et al. [50] reported that high-grain diets could result in SARA and LPS release. Ruminal-derived LPS could decrease the expression of tight junction proteins and impair rumen epithelial function. Another study demonstrated that increased histamine in ruminal fluid during SARA could disrupt ruminal epithelial barrier function [78,79].

### 4.3. Liver Dysfunction in SARA 

The liver is an important organ for human health and is required for metabolic activities, nutrient storage, detoxification and immunological activities. Harmful intestinal bacteria and metabolites pass through the intestine and need to be detoxified and filtered by the liver before they can reach other tissues or organs. Under normal circumstances, intestinal bacteria, bacterial metabolites and other harmful substances enter the liver through the blood and can be taken up and cleared by Kupffer cells in the liver [80]. Meanwhile, the liver can also chemically modify toxins, a variety of chemicals and even human metabolic waste through biotransformation reactions, turn these components into highly water-soluble components and then excrete them through the kidney. However, when the gut microbiota is dysbiotic, gut permeability increases, and bacteria and metabolites enter the liver through the blood, exceeding the detoxification and filtration capacity of the liver and then entering the blood and distant tissues, resulting in related diseases [81]. At the same time, these bacteria and metabolites will also cause a liver inflammatory response and damage, affecting the detoxification ability of the liver and resulting in more toxins entering the blood. Therefore, does liver dysfunction occur in the pathogenesis of SARA? After detoxification and filtration into the liver, which substances can enter the body or other organs through the blood or other ways, causing the occurrence of related diseases? 

Combined with the literature, we found that liver dysfunction occurred in the pathogenesis of SARA [2]. Tsuchiya et al. [14] reported that increased aspartate transaminase (AST) and non-esterified fatty acid (NEFA) in the blood were observed in Holstein cows with SARA. Recent studies showed that a high-concentrate diet could result in liver pathological injury, such as inflammatory cell infiltration and hepatocyte swelling and degeneration, suggesting that SARA could lead to liver injury [24]. Furthermore, Chang et al. [82] reported that high-concentrate, diet-induced SARA caused hepatocyte apoptosis by activating the extrinsic apoptosis pathway. Dong et al. [83] showed that increased expression of IL-1β, serum amyloid A, C-reactive protein and haptoglobin in liver tissues was observed in lactating goats with high-concentrate, diet-induced SARA. In addition, studies have shown that rumen-derived LPS and d-glutamyl-meso-diaminopimelic acid (iE-DAP) can induce liver inflammatory injury [84]. These results indicated that liver dysfunction and liver injury occurred in the pathogenesis of SARA. 

### 4.4. Ruminal Microbiota Dysbiosis Leads to the Release of Metabolites into the Blood and Tissues, Causing Inflammation and Related Diseases

#### 4.4.1. LPS

Under normal physiological conditions, the rumen and blood contain a small amount of LPS, which is not harmful to animal health. However, the occurrence of SARA will lead to a decrease in the pH value in the rumen, which results in the death of a large number of Gram-negative bacteria and the release of a large amount of LPS. Plaizier et al. [85] reported that the concentration of LPS in the blood, rumen, ileum, caecum and feces of SARA animals increased by 20-, 35-, 27.5-, 7- and 7-fold, respectively. Studies have shown that high-grain, diet-induced SARA could lead to the shift of LPS to the peripheral circulation and trigger a systemic inflammatory response [49]. When LPS is released into the blood, it can induce the production of haptoglobin (HP), serum amyloid A (SAA), LBP and the inflammatory cytokines TNF-α and IL-1β [5,49]. Furthermore, a large body of research has demonstrated that LPS can enter other tissues or organs through the blood and lead to related diseases [86]. Zhao et al. [87] showed that SARA resulted in high concentrations of rumen LPS, which activated the NF-κB and MAPK signaling pathways and led to the release of inflammatory cytokines in the rumen epithelium, causing rumenitis. Guo et al. [23] found that in a high-grain, diet-induced SARA model, rumen-derived LPS was transported from the digestive tract to the liver through the portal vein, causing hepatocyte injury, liver dysfunction and a liver inflammatory response. Bilal et al. [32] reported that LPS derived from the digestive tract could enter into the uterus and activate the TLR4 signaling pathway, causing endometritis. In our study, we also found that increased LPS in SARA could increase the number of milk somatic cells, causing mastitis. Zhang et al. [37] reported that increased LPS in SARA could activate the inflammatory response in lamellar tissues, which may progress to the level of laminar damage.

#### 4.4.2. Histamine

Histamine is an important mediator of type I allergies, which can cause vasodilation and increase vascular permeability. Meanwhile, it is also an important inflammatory mediator and has the ability to cause immune injury to humans and animals. Histamine is synthesized by decarboxylation of histidine via L-histidine decarboxylase [88]. *Allisonella histaminiformans* is the major producer of histamine in the rumen, and histidine is the only energy source for this bacterium [89]. It is generally believed that bacteria producing decarboxylase in the rumen can tolerate acidic environments. In SARA, the pH in the rumen decreases, which leads to dysbiosis of the rumen microbiota and increases in histamine-producing bacteria, finally resulting in an increased histamine concentration in the rumen [90,91]. Under normal physiological conditions, histamine is present in all tissues in trace amounts. The histamine entering the body is generally rapidly transformed into inactive substances through methylation or oxidation in the liver and excreted from the urine. When the histamine content increases beyond the body’s metabolic capacity, it enters the blood and tissues, causing systemic inflammation. Sun et al. [79] found that histamine could activate the NF-κB signaling pathway and inflammatory cytokine production, which subsequently led to the injury of bovine rumen epithelial cells. Previous studies showed that the increase in histamine was also closely related to mammary inflammation and lactation function [90]. They found that histamine could activate the NF-κB and mTOR signaling pathways, resulting in mammary inflammation and casein synthesis reduction [90]. Furthermore, studies have shown that histamine plays a critical role in the pathogenesis of laminitis [92]. When cattle are fed large amounts of grain, histamine can accumulate in the rumen and cause acute inflammation of the hooves (laminitis) [93].

#### 4.4.3. Other Metabolites 

In addition to LPS and histamine, other metabolites in the rumen can also enter the blood and cause or promote the occurrence of some diseases. γ-D-Glutamyl-meso-diaminopimelic acid (iE-DAP), which constitutes the peptidoglycan (PGN) layer of bacteria, increased significantly in the rumen fluid and blood of dairy cows suffering from SARA [94]. Studies have shown that iE-DAP can activate the NOD1-NF-κB signaling pathway and induce inflammation and injury in bovine hepatocytes and mammary epithelial cells [95]. Lactate, particularly D-lactate, increased significantly in the rumen in cases of SARA and high-grain, diet-induced SARA. Previous studies have shown that the increase in D-lactate is closely related to the occurrence of laminitis [96]. A previous study demonstrated that D-lactate could induce inflammation in bovine fibroblast-like synoviocytes by activating the MAPK and NF-κB signaling pathways [97]. Furthermore, the intraruminal injection of lactic acid could induce the occurrence of laminitis in lambs [98]. In addition to the above metabolites, there are many metabolites that increase in the pathogenesis of SARA. The role of these metabolites needs to be further investigated. Meanwhile, some metabolites were decreased in the pathogenesis of SARA, such as cholic acid, L-ascorbic acid, adenine, 2-hydroxyvaleric acid, 3-hydroxysebacic acid, 5′-methythioadenosine, deoxyribose, hippuric acid, phosphorycholine, L-isoleucine, 3-hydroxytetradecanedipnic acid and propenoycarnitine and so on [99]. These metabolites may have anti-inflammatory, tissue barrier protective effects or other effects. These metabolites may have potential for the treatment of SARA and its related diseases. 

### 4.5. Ruminal Microbiota Dysbiosis Leads to Bacterial Translocation, Causing Inflammation and Related Diseases

Bacterial translocation is the process by which gut bacteria cross the intestinal mucosal barrier into the mesenteric lymph nodes and portal system and then into systemic circulation and organs [100]. Just as the translocation of endotoxins into the blood causes an inflammatory response in the body, bacterial translocation can also cause systemic inflammation or damage to distant organs. Previous studies have shown that the intestines were the main source of endogenous bacteria causing infections in a variety of diseases [101,102]. Previous studies have shown that the course of acute pancreatitis may be accompanied by pancreatic infection caused by bacterial migration, mainly due to the loss of intestinal barrier function during the disease process [103]. In addition, breast milk was once thought to be sterile, but more recent studies have shown that breast milk not only contains various types of bacteria but also originates from the mother’s gut [104]. Furthermore, bacterial translocation due to impaired intestinal epithelial barrier integrity also exists in end-stage renal disease, which plays an important role in disease progression [105]. Similar to the above diseases, bacterial translocation is also present in SARA. This is mainly because SARA is accompanied by dysbiosis of the rumen microbiota, which induces impairment of the intestinal barrier, resulting in bacterial translocation.

As mentioned above, the occurrence of SARA is also accompanied by bacterial translocation. Jeon et al. [106] showed that pathogenic bacteria could migrate from the gut to the uterus through the blood, causing metritis in dairy cows. In high-grain, diet-induced SARA, we found that *Stenotrophomonas* was enriched in the rumen and appeared in the mammary gland. Meanwhile, we found that *Stenotrophomonas* could cause mastitis in mice by gavage. The results indicate that the massively proliferating *Stenotrophomonas* in the rumen may translocate to the mammary gland through a certain pathway, leading to the occurrence of mastitis [2]. Whether the bacteria in the rumen can migrate to other tissues or organs and through which mechanisms still need to be further studied. 

### 4.6. Ruminal Microbiota Dysbiosis Facilitates Susceptibility to Pathogens 

Rumen microorganisms are necessary “organs” for normal physiological homeostasis and affect the immune system’s response to pathogens. Studies have shown that disruption of the homogeneous gastrointestinal microbiota may thus facilitate the development of pathogen-induced diseases. Accumulating studies have demonstrated that the gut microbiota is involved in the maturation of the immune system. It stimulates innate immunity in the early years of life, leading to the maturation of gut-associated lymphoid tissue and acquired immunity through the stimulation of local and systemic immune responses [107]. The commensal gastrointestinal microbiome competitively restricts pathogen survival and proliferation, by metabolizing and consuming nutrients on one side and by producing inhibitory molecules on the other [101]. The gastrointestinal microbiota coevolved with the host’s immune system and established a delicate balance to maintain homeostasis in the gut [108]. These microorganisms have a fundamental effect on regulating immune cell development, differentiation and defense against pathogen invasion [109]. In recent years, germ-free animals have been used to study the importance of the gut microbiota for the development of the innate immune system. Relative to conventionally reared mice, germ-free mice have thinner gut tissue, a smaller mucus layer, sparse immune cells in the lamina propria and smaller Peyer’s patches [110,111]. These changes in germ-free mice undoubtedly lead to weakened intestinal immunity and increased pathogen susceptibility. In addition, germ-free mice have severely immune-deficient immune systems, such as lower levels of immunoglobulin A (IgA) and IL-17-producing T helper 17 cells in their gut, but these phenomena can be reversed by microbiota supplementation [112]. At the systemic level, they show decreased antibody production, lower plasma levels and smaller mesenteric lymph nodes, suggesting that the absence of the microbiota affects the systemic immune response [110,113]. Overall, alterations in the composition of the microbial community rearrange the structure of the immune system and impair the resistance of the host’s immune system to pathogens. In ruminants, studies have shown that rumen microbiota dysbiosis can affect metabolites and decrease the immune responses of sheep under cold stimulation. Increased rumen microbiota dysbiosis, especially SARA, is associated with pathogen-induced diseases [114]. Our previous results showed that cows with SARA caused by high-grain-diet consumption had increased susceptibility to *S. aureus*-induced mastitis (data have not been published). In addition, SARA cows treated with LPS through intramammary infusion had higher acute phase proteins and modulated the blood metabolome differently than LPS-treated cows without SARA [115]. These studies suggested that rumen microbiota dysbiosis could facilitate susceptibility to pathogens. The mechanism still needs to be further studied. 

## 5. Regulating the Rumen Microbiota to Prevent SARA and Related Diseases in Dairy Cows

Disturbances of the rumen microbiota and metabolites are closely related to the occurrence of SARA and related diseases. Therefore, targeted regulation of the rumen microbiota may be an important approach to prevent and treat SARA and related diseases. Evidence showed that thiamine supplementation inhibited gut injury, increased the relative abundance of beneficial bacteria and reduced intestinal dysbacteriosis in SARA cows induced by a high-concentrate diet [116]. Others also indicated that plant-derived extracts, including alkaloids, terpenoids and essential oils, could maintain ruminal pH and improve ruminal fermentation [117]. The release of LPS from the gastrointestinal microbiota into the blood is one of the important factors for SARA to increase other metabolic and infectious diseases. Evidence showed that treatment with an anti-LPS antibody reduced the release of LPS and inhibited pH in the rumen in cows suffering from SARA [118,119]. These results indicate that anti-LPS antibodies may be used for the prevention and treatment of metabolic and infectious diseases related to rumen flora disturbance in dairy cows. 

In addition, rumen transplantation is commonly used in the treatment of digestive disorders, including left-sided abomasal displacement [120]. Recently, evidence also suggested that rumen content transplantation (RCT) can return to normal levels with rumen fermentation parameters, including the levels of bacterial community diversity and the concentrations of acetate, valerate and VFA, in SARA cows induced by a high-grain diet [65]. Furthermore, rumen fluid transplantation (RT) also changed the gastrointestinal microbiota and then influenced the feed intake, feed digestibility and growth performance of weaned lambs [121]. Others also demonstrated that RT restored ruminal bacterial homeostasis, increased the concentrations of VFA, acetate, propionate and butyrate and decreased the concentrations of lactate and LPS in the rumen [122]. Although there is still a lack of studies on the effect of rumen flora transplantation on inflammatory diseases of the distal extra-gut organs in dairy cows, ours and other studies found that rumen flora and metabolites disturbance can induce mastitis, endometritis and laminitis in dairy cows [2,32]. RCT can effectively relieve SARA. Therefore, we speculate that rumen flora transplantation may be used for the prevention and treatment of inflammatory diseases associated with SARA in ruminants. 

## 6. Conclusions

SARA is an important metabolic disorder in dairy cows that affects animal welfare and the economy of milk production. It often leads to local inflammatory diseases such as laminitis, mastitis, endometritis and hepatitis. Clarifying the mechanism of SARA-mediated inflammatory diseases is of great significance for the joint prevention and control of these diseases. This review concludes that the rumen microbiota and its metabolites play a critical role in SARA-mediated inflammatory diseases. It is a bridge of SARA and its related inflammatory diseases. Dysbiosis of the rumen microbiota, destruction of the rumen barrier and dysbiosis of liver function in the pathogenesis of SARA will lead to the entry of rumen bacteria and/or metabolites into the body and place the body in the chronic low-grade inflammatory state. Meanwhile, rumen bacteria and/or their metabolites can also migrate to the mammary gland, uterus and other organs, leading to the occurrence of related inflammatory diseases. The rumen microbiota can be used as a target for the treatment of SARA and its related diseases. 

## Figures and Tables

**Figure 1 microorganisms-10-01495-f001:**
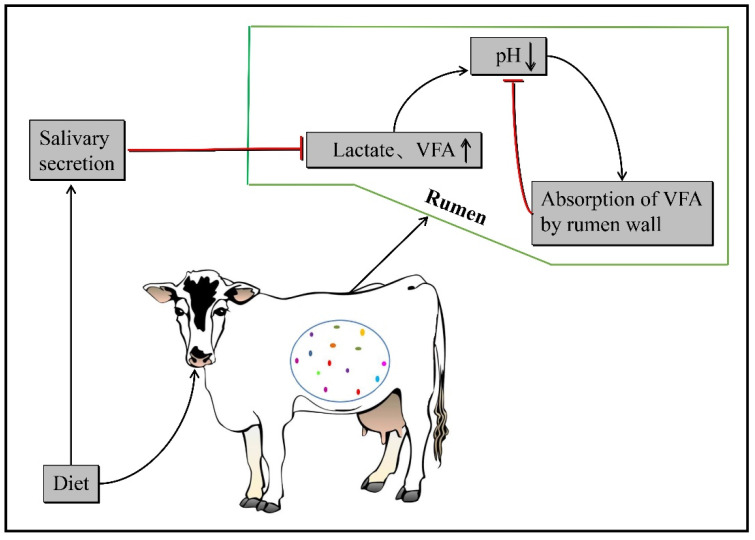
Physiology of the fermentation process in the rumen of dairy cows.

**Figure 2 microorganisms-10-01495-f002:**
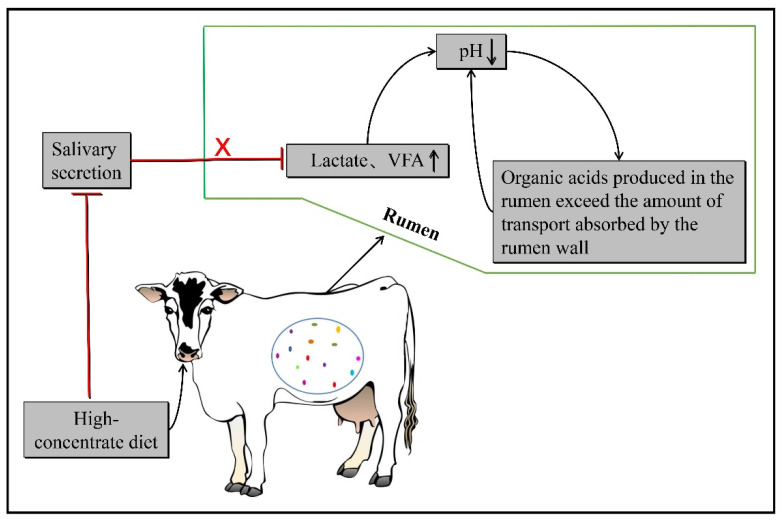
Pathogenesis of SARA in dairy cows.

**Figure 3 microorganisms-10-01495-f003:**
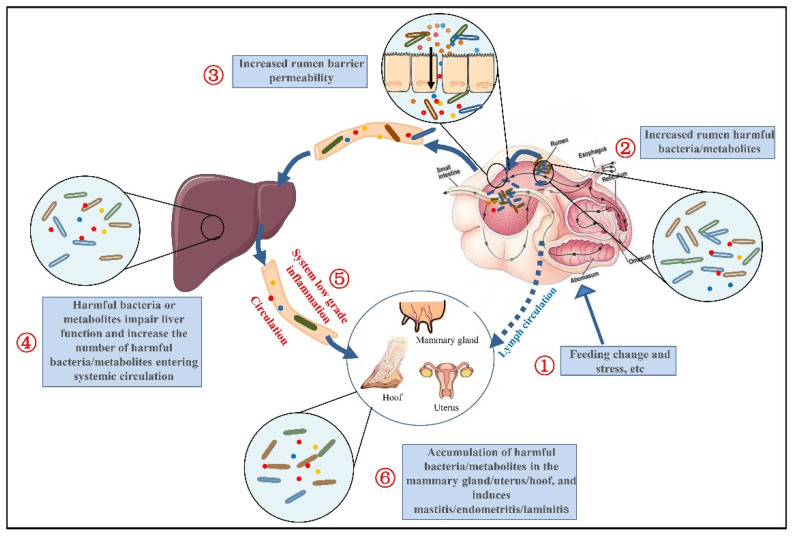
Mechanism by which SARA increases infectious and metabolic diseases in dairy cows.

**Table 1 microorganisms-10-01495-t001:** The changes of inflammatory markers in animals with SARA.

Inflammatory Biomarkers	Animal Species/Test Samples	Disease Group Animals	Control Group Animals
LPS	Goat/Cecal contents	19,889.47 ^a^ ± 2917.37 EU/mL	7257.01 ± 1020.43 EU/mL [47]
	Cattle/Rumen fluid	51,481 ^a^ EU/mL	13,331 EU/mL [48]
	Dairy cows/Rumen fluid	151,985 ^a^ EU mL	29,492 EU/mL [49]
	Dairy cows/Peripheral blood	0.81 ^a^ EU/mL	<0.05 EU/mL [49]
	Dairy cows/Rumen fluid	89.3 ^a^ kEU/mL	34.2 kEU/mL [50]
	Dairy cows/Peripheral blood	0.37 ^a^ EU/mL	0.16 EU/mL [50]
	Dairy cows/Rumen fluid	78.43 ^a^ kEU/mL	47.47 kEU/mL [51]
	Dairy cows/lacteal artery plasma	0.85 ^a^ EU/mL	0.45 EU/mL [51]
	Dairy cows/lacteal vein plasma	0.25 ^a^ EU/mL	0.15 EU/mL [51]
	Dairy cows/Feces	252,345 ^a^ EU/g	3514 EU/g [51]
Histamine	Dairy cows/Rumen fluid	64 ^a^ μmol/L	0.5 μmol/L [52]
	Dairy cows/Peripheral blood	0.2 ^a^ μmol/L	<0.009 μmol/L [52]
	Dairy cows/Rumen fluid	161.2 ** μmol/L	46.4 μmol/L [53]
	Dairy cows/Peripheral blood	7.92 ** μmol/L	2.03 μmol/L [53]
TNF-α	Dairy cows/Peripheral blood	18.56 ^a^ fmol/mL	9.83 fmol/mL [50]
IL-1β	Dairy cows/Peripheral blood	1.07 ^a^ ng/mL	0.32 ng/mL [50]
IL-6	Dairy cows/Peripheral blood	532.18 ^a^ pg/mL	98.36 pg/mL [50]
SAA	Dairy cows/Peripheral blood	446.7 ^a^ μg/mL	164.4 μg/mL [49]
	Dairy cows/Peripheral blood	498.8 ^a^ μg/mL	286.8 μg/mL [54]
	Dairy cows/Peripheral blood	170.7 ^a^ ± 36.53μg/mL	33.6 ± 36.53 μg/mL [5]
Hp	Dairy cows/Peripheral blood	484 ^a^ μg/mL	<50 μg/mL [49]
	Dairy cows/Peripheral blood	265 ^a^ μg/mL	244 μg/mL [54]
	Beef cattle/Peripheral blood	0.79 ^a^ ± 0.14 mg/mL	0.43 ± 0.14 mg/mL [5]
LBP	Dairy cows/Peripheral blood	53.1 ^a^ μg/mL	18.2 μg/mL [49]
	Dairy cows/Milk	6.94 ^a^ μg/mL	3.02 μg/mL [49]
WBC	Dairy cows/Peripheral blood	5.69 × 10^9^/L	5.23 × 10^9^/L [54]

Note: ^a^ represents significant difference compared with the control group; ** represents extremely significant difference compared with the control group.

## Data Availability

Not applicable.

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
