# Peer review of "The Role of Rumen Microbiota and Its Metabolites in Subacute Ruminal Acidosis (SARA)-Induced Inflammatory Diseases of Ruminants"

_microorganisms, 2022, doi:10.3390/microorganisms10081495_

Round 1

Reviewer 1 Report

The review article written by Yunhe Fu, et al. reviewed the "Rumen microbiota and its metabolites: Bridge of subacute ruminal acidosis (SARA) and inflammatory diseases of ruminant". The manuscript was exciting to read and presents interesting information about rumen microbiota. However, I did not recommend this article for publication.

  • Microorganism The templated manuscript is missing. It is more complex to understand.
  • Some mistakes have been detected. For example, figure captions are confusing (line no: 82-85)
  • There are a lot of grammatical mistakes in this manuscript.
  • The author did not discuss metabolites from rumen microbiota disturbance.

  • There are no scientific sounds in Figure 1 and 2. The author could improve more.

Author Response

Response to reviewers

Thank you for your letter and for the reviewers’ comments concerning our manuscript entitled “Rumen microbiota and its metabolites: Bridge of subacute ruminal acidosis (SARA) and inflammatory diseases of ruminant” (microorganisms-1790094). These comments are all valuable and very helpful for revising and improving our paper, as well as the important guiding significance to our researches. We have studied comments carefully and have made corrections which we hope meet with approval.

Point-by-point response

The review article written by Yunhe Fu, et al. reviewed the "Rumen microbiota and its metabolites: Bridge of subacute ruminal acidosis (SARA) and inflammatory diseases of ruminant". The manuscript was exciting to read and presents interesting information about rumen microbiota. However, I did not recommend this article for publication.

Microorganism The templated manuscript is missing. It is more complex to understand.

ResponseWe have rewritten the manuscript as requested. Thank you!

Some mistakes have been detected. For example, figure captions are confusing (line no: 82-85)

Response: We have depleted the sentence in line 82-85, and we have added the figure legends in figure 1 to figure 3. Thank you!

There are a lot of grammatical mistakes in this manuscript.

Response We have touched up the manuscript.

The author did not discuss metabolites from rumen microbiota disturbance.

Response: We have added the metabolites from rumen microbiota disturbance in the line 253-381. Thank you!

Reviewer 2 Report

This is an interesting review, but, a review of the english language used and grammar by a native speaker would improve it. 

A few minor points, 

The title is not really clear 

The figures need to be clearer and have a figure legend. 

Did the authors consider discussing treatments aimed at the microbiome? 

Author Response

Response to reviewers

Thank you for your letter and for the reviewers’ comments concerning our manuscript entitled “Rumen microbiota and its metabolites: Bridge of subacute ruminal acidosis (SARA) and inflammatory diseases of ruminant” (microorganisms-1790094). These comments are all valuable and very helpful for revising and improving our paper, as well as the important guiding significance to our researches. We have studied comments carefully and have made corrections which we hope meet with approval.

Point-by-point response

This is an interesting review, but, a review of the english language used and grammar by a native speaker would improve it. 

A few minor points, 

The title is not really clear 

ResponseWe have changed the title to “The role of rumen microbiota and its metabolites in subacute ruminal acidosis (SARA)-induced inflammatory diseases of ruminants”. Thank you!

The figures need to be clearer and have a figure legend. 

ResponseWe have changed the figures and added the figure legends from figure 1 to figure 3. Thank you!

Did the authors consider discussing treatments aimed at the microbiome? 

ResponseWe have added the discussing treatments aimed at the microbiome in lines 442-470. Thank you!

Round 2

Reviewer 1 Report

None